# Density Functional Theory Calculations of the Effect of Oxygenated Functionals on Activated Carbon towards Cresol Adsorption

**Aola Supong, Upasana Bora Sinha and Dipak Sinha \***

Department of Chemistry, Nagaland University, Lumami, Zunheboto 798627, India; aolas@rediffmail.com (A.S.); upasanaborasinha@gmail.com (U.B.S.)
* Correspondence: dipaksinha@gmail.com

**Abstract:** The mechanism of adsorption of *p*-cresol over activated carbon adsorbent and the specific role of oxygen functional groups on cresol adsorption were studied using density functional theory (DFT) calculations. All the energy calculations and geometry optimization pertaining to DFT calculations were done using the B3LYP hybrid functional at basis set 6-31g level of theory in a dielectric medium of $\varepsilon = 80$ (corresponding to water). The interaction of cresol with different activated carbon models, namely pristine activated carbon, hydroxyl functionalized activated carbon, carbonyl functionalized activated carbon, and carboxyl functionalized activated carbon, were considered, and their adsorption energies corresponded to −416.47 kJ/mol, −54.73 kJ/mol, −49.99 kJ/mol, and −63.62 kJ/mol, respectively. The high adsorption energies suggested the chemisorptive nature of the cresol-activated carbon adsorption process. Among the oxygen functional groups, the carboxyl group tended to influence the adsorption process more than the hydroxyl and carbonyl groups, attributing to the formation of two types of hydrogen bonds between the carboxyl activated carbon and the cresol simultaneously. The outcomes of this study may provide valuable insights for future directions to design activated carbon with improved performance towards cresol adsorption.

**Keywords:** activated carbon; DFT; cresol

## 1. Introduction

Adsorption technology is one of the modern technologies which has been increasingly used for combating various environmental issues, particularly to overcome the degradation of water quality. Among various adsorbents used in different adsorption processes, activated carbon is one of the most common and effective adsorbents owing to its adequate porous nature, high surface area, large adsorption capacity, versatility, and high surface reactivity [1,2]. By virtue of these attractive characteristics, activated carbon is considered a unique and versatile carbon material that has been widely used for the treatment of groundwater contaminated with volatile organics, deteriorated drinking water sources, removing pollutants from wastewater, etc. [3–8]. The presence of synthetic organic pollutants in water resources has become a serious environmental issue in recent decades. The contamination of water bodies with such pollutants is a direct consequence of the rapid development of industrial activities [9]. Among the host of organic pollutants discharged in industrial effluents, phenolic compounds are major contaminant compounds and the Environmental Protection Agency (EPA) has categorized them as priority pollutants because of their high toxicity and low biodegradability [10,11]. *p*-cresol (4-methyl phenol) is one of the highly toxic derivatives of phenol and poses a significant risk to both humans and the environment. It is frequently detected in effluents generated from various chemical and allied industries such as petrochemicals, oil refineries, pulp and paper mill industries, steel plants, ceramic plants, coking plants, polymeric resin manufacturing, coal conversion,

and phenol-producing industries. The toxic effects of *p*-cresol on human skin, eyes, respiratory tract, and nervous system can be perceived even at very minute concentrations. It can cause severe damage to the heart, liver, kidneys, and nerve cells [12]. In addition, *p*-cresol has been classified as a possible carcinogen by the EPA [13]. Hence, with the ever-increasing threat of *p*-cresol to public health and environmental quality, the mitigation of such pollutants from water bodies remains a key challenge.

An adsorbate's adsorption on activated carbon largely depends on the chemical and physical features of the activated carbon [14]. The physical features are attributed to the pore structure and surface area of the activated carbon, while the chemical characteristics are largely determined by the functionals on the activated carbon surface [15]. Among the various functional groups, oxygen-containing functional groups are by far the most common and abundant groups [16–19] ]. These functional groups are either derived from the raw material during carbonization or introduced during the activation process [20,21]. The amount and nature of the functional groups largely depend on the nature of the carbon surface, its surface area, and the activation conditions [21]. Carboxyl, hydroxyl, carbonyl, and ketone groups are among the most abundant oxygen-containing functional groups, and such oxygenated functional groups have been found to significantly affect the adsorptive nature of the activated carbon [1,22–24]

Although numerous experimental works have been carried out on the utilization of activated carbon for the removal of cresols [25–27], studies to understand the adsorption mechanisms and also the effect of oxygenated functionals on the adsorption process are very limited. Therefore, an in-depth study of the interaction between cresol and pristine/functionalized activated carbon would provide further insights into the adsorption process. In present day research, theoretical DFT calculations are considered as one of the most powerful tools for predicting interaction mechanisms between adsorbents and adsorbates in surface science [28]. In this regard, various studies have been conducted to understand the adsorption mechanisms of phenols on activated carbon surfaces by employing DFT calculations. For instance, Cam et al. [29] reported that the aromatic part of phenol interacts with the basal planes of activated carbon through weak physical adsorption. Their results also suggested that the carboxylic group interacted most strongly with the phenol. Yin et al. [30] suggested that the phenol uptake on activated carbon decreased with an increase in the concentration of oxygenated functional groups. The affinity of different oxygen functional groups on activated carbon has also been studied by DFT calculations considering the hydrogen bond lengths and the corresponding thermodynamic parameters [31]. Liu et al. also studied the effect of oxygen and nitrogen functional groups on phenol adsorption, wherein they reported the high affinity of carboxyl and pyridine groups for phenol adsorption [32]. A DFT study of phenol adsorption on pristine and functionalized graphene for phenol removal studies was reported by Ghahghaey et al.; their study suggested the high adsorption capacity of hydroxyl-graphene as compared to pristine-graphene and other counterparts [33]. Our previous studies relating to theoretical studies of the interaction between activated carbon and different types of phenolic compounds such as phenol, bisphenol A, 4-nitrophenol, and 2,4-dinitrophenol provided different interaction routes and also explained the positive effect of oxygenated functionals on the adsorbate–adsorbent interaction [34–36]. In addition to phenolic compounds, DFT studies have also been successful in incorporating the study of the interaction of activated carbon with other compounds, such as Congo red dye [37], paracetamol [38], atrazine [37], azo dye [39], etc. However, to date, almost no studies have been reported on the interaction between cresol and activated carbon, even though cresol is an important pollutant which is quite commonly encountered. Thus, keeping in mind the accuracy and simplicity of DFT calculations and the insights obtained in our previous works, the present study was initiated which employed DFT calculations to study the interaction between activated carbon and *p*-cresol. In recent years, density functional theory (DFT) calculation has been widely adopted to study the interacting mechanisms which are difficult to understand experimentally, and, in this study, the DFT calculation was carried out to ascertain the

interacting behaviors of cresol adsorption on pristine and oxygen functional group surfaces of activated carbon. Different types of interactive mechanisms were explored, and the effect of three major functionals (hydroxyl, carboxyl, and carbonyl) on the *p*-cresol adsorption was studied.

The novelty of the present work lies in the use of DFT calculations for understanding the interaction of cresol with pristine and functionalized activated carbon. The study provides new findings on the effect of different oxygenated functional groups, namely hydroxyl, carboxyl, and carbonyl, on cresol adsorption. Although several studies have been reported on the adsorption of cresol on activated carbon, to date, however, almost no studies have been reported on the interaction mechanism. The present study thus provides a new understanding of the interaction between pristine activated carbon and functionalized activated carbon via hydrogen bond formation using DFT. The outcomes of this study may provide directions to design activated carbon with specialized characteristics.

## 2. Materials and Methods

Density functional theory (DFT) calculations were performed using the Gaussian 09 program package [40]. The energy calculations and geometry optimization were done by using the B3LYP hybrid functional at basis set 6-31g level of theory in a dielectric medium of $\varepsilon = 80$ (corresponding to water). Gauss View 05 generated all the structures, which were fully optimized to their electronic ground state. To study the interactions of activated carbon with *p*-cresol, it was necessary to first create an appropriate model for the activated carbon. According to solid-state $^{13}$C NMR experiments, the activated carbon surface consists of 3 to 7 fused benzene rings [41]. As such, previous studies have used carbon models consisting of 4 to 7 fused benzene rings to represent the activated carbon surface, which has proved to be important in comprehending the interaction of different adsorbates with the activated carbon surface [42–45]. Thus, in the present work, an arm-chair model was used to represent the activated carbon surface. This model consisted of four fused benzene rings where the upper edge atoms were unsaturated to represent the active site, and the remaining carbon atoms on the lower side were bonded with hydrogen atoms. This model with an active site-unsaturated carbon was considered as pristine activated carbon. The influence of oxygenated functionals on the adsorption process was studied by considering three oxygen-containing functional groups, namely carboxyl, hydroxyl, and carbonyl, where each functional was bonded to the active site of the arm-chair model. Figure 1 represents the optimized structure of *p*-cresol and all the activated carbon models. The models were named pristine activated carbon (AC), hydroxyl functionalized activated carbon (AC-OH), carbonyl functionalized activated carbon (AC-CHO), and carboxyl functionalized activated carbon (AC-COOH).

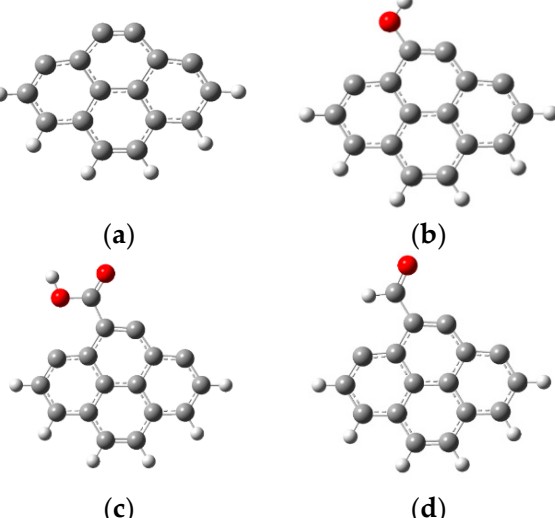

(a)  (b)

(c)  (d)

**Figure 1.** *Cont.*

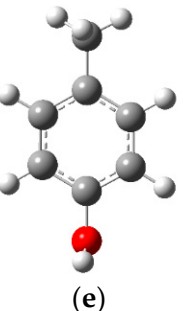

(**e**)

**Figure 1.** (**a**) AC, (**b**) AC-OH, (**c**) AC-COOH, (**d**) AC-CHO, and (**e**) *p*-cresol optimized structures.

The adsorption energy of *p*-cresol with the pristine and functionalized carbon models was calculated by the equation

$$E_{AC} = E_{cresol+AC} - (E_{cresol} + E_{AC})$$

where $E_{cresol+AC}$ is the total energy of the cresol and activated carbon system; $E_{AC}$ is the total energy of the activated carbon, and $E_{cresol}$ is the total energy of the *p*-cresol. Generally, adsorption energy of less than −30 kJ/mol and more than −50 kJ/mol indicates physisorption and chemisorption, respectively. A higher negative value of adsorption energy indicates stronger adsorption [45].

### 3. Results

*3.1. Cresol Adsorption on Activated Carbon*

The adsorption of cresol on activated carbon is expected to proceed through various types of interactions such as electron donor–acceptor mechanisms, electrostatic interactions, π–π interactions, and hydrogen bond formation [46]. However, among the various types of interactions, hydrogen bonding between the activated carbon and adsorbate is one of the important types of interactions contributing to the adsorption process. Thus, in this study, hydrogen bond interactions between *p*-cresol and activated carbon were studied using the DFT approach. The interaction of *p*-cresol with pristine activated carbon and the functionalized activated carbon was studied considering the hydrogen bonding between them. Figure 2 represents the optimized structures of the various interactions, whereas the bond distance and the adsorption energies are given in Table 1.

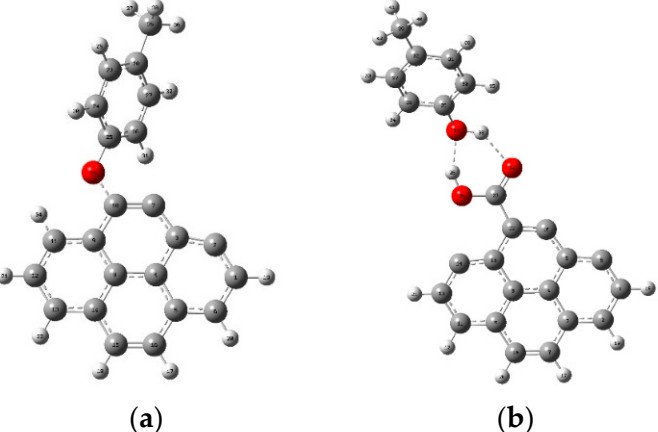

(**a**) (**b**)

**Figure 2.** *Cont.*

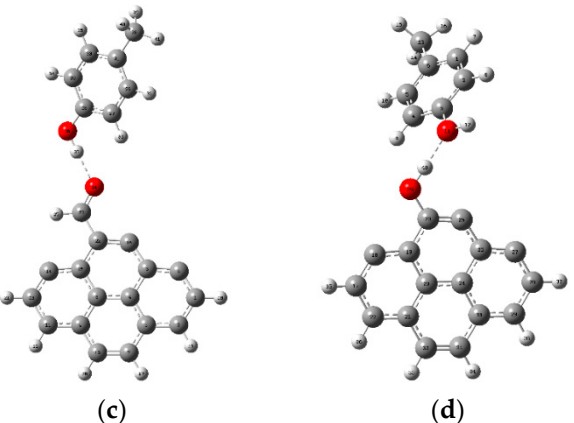

<div align="center">(<b>c</b>)                     (<b>d</b>)</div>

**Figure 2.** Optimized structures representing (**a**) $C_7H_7$-OH—-C-AC, (**b**) $C_7H_7$-HO—-HOOC-AC, (**c**) $C_7H_7$-OH—OHC-AC, and (**d**) $C_7H_7$-HO—-HO-AC interactions.

**Table 1.** Bond distance and adsorption energy of the adsorbate–adsorbent interaction.

| System | Mode of Interaction | Adsorption Energy (kJ/mol) | Bond Length (Å) |
|---|---|---|---|
| $C_7H_7$-OH + AC | $C_7H_7$-OH—-C-AC | −416.47 | 1.40 ($O_{cresol}$-$C_{AC}$) 1.08 ($H_{cresol}$-$C_{AC}$) |
| $C_7H_7$-OH + AC-OH | $C_7H_7$-OH—-HO-AC | −54.73 | 1.65 ($O_{cresol}$-$H_{AC-OH}$) |
| $C_7H_7$-OH + AC-CHO | $C_7H_7$-OH—-OHC-AC | −49.99 | 1.72 ($H_{cresol}$-$O_{AC-COOH}$) |
| $C_7H_7$-OH + AC-COOH | $C_7H_7$-OH—-HOOC-AC | −63.62 | 1.87 ($H_{cresol}$-$O_{AC-COOH}$) 1.68 ($O_{cresol}$-$H_{AC-COOH}$) |

### 3.2. Theoretical Calculations

3.2.1. Cresol Adsorption on Pristine Activated Carbon

The adsorption of *p*-cresol ($C_7H_7$-OH) on pristine activated carbon (AC) system was studied by considering the $C_7H_7$-OH—-C-AC mode of interactions, in which the –OH group of *p*-cresol was oriented towards the carbon atom of the arm-chair edge site. Figure 2a represents the optimized structure of the cresol–pristine activated carbon interaction. The interaction was found to proceed via the dissociation of the $C_7H_7$-OH into $C_7H_7$-O and H, resulting in the formation of $C_7H_7$-O—-C-AC and $H_{cresol}$—-C-AC types of interactions. The favorability of the $C_7H_7$-OH—-C-AC type of interactions was also indicated by the elongation of the C–C bond distance of the pristine activated carbon where $C_7H_7$-O and H were directly attached. The C–C bond distance increased from 1.24 Å to 1.34 Å upon its interaction with *p*-cresol. These results indicate that the C–C bond weakens as a result of the shifting of the electron cloud towards the adsorption site, i.e., $C_7H_7$-O—-C-AC bond. The bond distance of $C_7H_7$-O—-C-AC and $H_{cresol}$—-C-AC corresponded to 1.40 Å and 1.08 Å, respectively. Moreover, the adsorption energy of the $C_7H_7$-OH—-C-AC was −416.473 kJ/mol indicating a thermodynamically favorable chemisorptive type of adsorption process.

3.2.2. Cresol Adsorption on –COOH Functionalized Activated Carbon

The adsorption of *p*-cresol with the carboxyl functionalized activated carbon took place through the bonding via two hydrogen bonds between cresol and AC-COOH. One of the hydrogen bonds was formed between the hydrogen atom of AC-COOH and the oxygen atom of cresol, while another hydrogen bond existed between the oxygen atom of AC-COOH and the hydroxyl hydrogen of cresol. The two types of hydrogen bonds are represented by $C_7H_7$-OH—-OHOC-AC and $C_7H_7$-HO—-HOOC-AC. Figure 2b represents the optimized interaction structure of the $C_7H_7$-OH and AC-COOH. Upon interaction of AC-COOH and cresol, the O–H bond length of both AC-COOH and cresol elongated from 0.98 Å to 1.01 Å and 0.97 Å to 0.99 Å, respectively. The bond distances of $C_7H_7$-OH—-

OHOC-AC and $C_7H_7$-HO—-HOOC-AC were 1.87 Å and 0.68 Å, respectively, whereas the adsorption energy was −63.62 kJ/mol. These observations indicated the favorability of the cresol interaction with the –COOH functionalized activated carbon.

### 3.2.3. Cresol Adsorption on –CHO Functionalized Activated Carbon

The adsorption of *p*-cresol with AC-CHO proceeded via the formation of the $C_7H_7$-OH—-OHC-AC type of interaction, in which the hydroxyl hydrogen atom of cresol interacted with the oxygen atom of AC-CHO. The optimized interaction structure is shown in Figure 2c. Upon interaction of cresol with AC-CHO, the O–H bond distance of cresol increased from 0.97 Å to 0.99 Å, while the C=O bond length of AC-CHO increased from 1.24 Å to 1.25 Å. These observations indicated that the O–H and C=O bonds become weaker because of the formation of a stronger $H_{cresol}$-$O_{AC-CHO}$ bond of the $C_7H_7$-OH—-OHC-AC interaction. The $H_{cresol}$-$O_{AC-CHO}$ bond distance was found to be 1.72 Å, while the adsorption energy was −49.99 kJ/mol. The short $H_{cresol}$-$O_{AC-COOH}$ bond length and the high negative energy indicated that the $C_7H_7$-OH—-OHC-AC type of interaction between cresol and AC-CHO was favorable.

### 3.2.4. Cresol Adsorption on –OH Functionalized Activated Carbon

The *p*-cresol interaction with –OH-functionalized activated carbon occurred favorably via the formation of a hydrogen bond between the hydroxyl hydrogen atom and the oxygen atom of cresol and AC-OH. The interaction is represented by $C_7H_7$-HO—-HO-AC. Figure 2d represents the optimized structure. Upon adsorption, the O–H bond of activated carbon elongated from 0.97 Å to 1 Å, indicating the shifting of the electron density towards the $O_{cresol}$-$H_{AC-OH}$ bond. This displacement of the electron cloud towards the adsorption site resulted in stronger $C_7H_7$-HO—-HO-AC interaction. The bond distance of $O_{cresol}$-$H_{AC-OH}$ was 1.65 Å, and the adsorption energy of the $C_7H_7$-HO—-HO-AC system was found to be −54.73 Å, indicating that the interaction proceeded via chemisorption.

### 3.3. Relative Adsorption Energies

A comparative study of the adsorption energies of the interaction of the *p*-cresol molecule with pristine activated carbon, AC-COOH, AC-OH, and AC-CHO was done to determine the best possible types of interaction for cresol adsorption. The energy profile diagram is given in Figure 3.

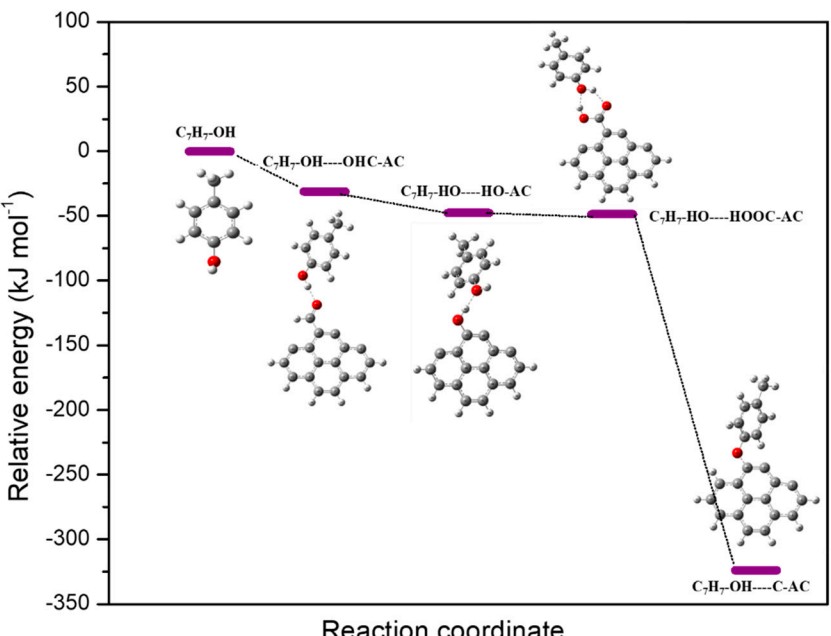

**Figure 3.** Relative energy diagram of adsorption of cresol onto pristine and functionalized activated carbon.

The adsorption energies of *p*-cresol with the pristine and functionalized activated carbon showed high negative energy, indicating that the interaction of cresol with both the pristine and functionalized activated carbon was favorable. However, the adsorption energy of cresol with pristine activated carbon was more negative as compared to functionalized activated carbon, indicating that the pristine activated carbon, i.e., activated carbon with a high degree of unsaturation, would be more favorable for optimum cresol adsorption.

Moreover, among the various functional groups considered for the present study, the activated carbon-containing –COOH group resulted in higher negative adsorption energy (63.62 kJ/mol) as compared to the –OH (−54.73 kJ/mol) and –CHO (−49.99 kJ/mol) groups. The formation of two hydrogen bonds between the cresol molecule and AC-COOH simultaneously compared to the formation of a single hydrogen bond with AC-CHO and AC-OH may have contributed to the stronger interaction. These theoretical findings may suggest that the introduction of the –COOH functionals to the activated carbon surface may increase the cresol interaction with activated carbon.

## 4. Discussion

The cresol adsorption on pristine and functionalized activated carbon was found to proceed favorably via the formation of hydrogen bonds. The adsorption energy of cresol with pristine activated carbon, hydroxyl functionalized activated carbon, carbonyl functionalized activated carbon, and carboxyl functionalized activated carbon corresponded to −416.47 kJ/mol, −54.73 kJ/mol, −49.99 kJ/mol, and −63.62 kJ/mol, respectively. The higher adsorption energy of pristine activated carbon may be attributed to the dissociative type of interaction of cresol with the pristine activated, wherein cresol $C_7H_7$-OH dissociates into $C_7H_7$-O and H, and both the dissociation parts form a strong bond with the carbon of the activated carbon surface. While in the case of oxygenated activated carbon surface cresol does not undergo dissociation, rather it reacts with the carbon surface as a complete moiety through the formation of a hydrogen bond. Among the different oxygenated functional groups, the activated carbon with the carboxyl group showed a stronger interaction, which may be due to the formation of double hydrogen bonds as compared to the single hydrogen bond formed between cresol and carbonyl or hydroxyl functionalized activated carbon. As seen in Figure 3, the cresol–pristine activated carbon occupies the lowest energy, which indicates the strongest interaction. The second lowest energy is showed by the carboxylic functionalized activated carbon–cresol interaction, indicating the favorability of the carboxyl group in cresol adsorption. The strength of the interaction of cresol with the different activated carbon proceeded in the following order:

$C_7H_7$-OH—-C-AC > $C_7H_7$-HO—-HOOC-AC > $C_7H_7$-HO—-HO-AC > $C_7H_7$-OH—-OHC-AC

## 5. Conclusions
- The DFT studies provided a new understanding of the interaction of *p*-cresol with pristine and functionalized activated carbon.
- The adsorption of cresol on pristine, hydroxyl functionalized activated carbon, carbonyl functionalized activated carbon, and carboxyl functionalized activated carbon were found to be favorable, and their adsorption energies corresponded to −416.47 kJ/mol, −54.73 kJ/mol, −49.99 kJ/mol, and −63.62 kJ/mol, respectively.
- The high adsorption energies suggested the chemisorptive type of interaction between cresol and activated carbon.
- The pristine activated carbon showed stronger adsorption towards cresol as compared to functionalized activated carbon, while among the oxygen functionals, the activated carbon-containing carboxyl interacted more favorably with the cresol compared to the hydroxyl and carbonyl group, attributed to the formation of two types of hydrogen bonds between the carboxyl activated carbon and the cresol simultaneously.
- The study indicates that the introduction of carboxyl functionals on the surface of activated carbon would favor cresol adsorption.

- The outcome of these theoretical findings could provide valuable guidance for the development and production of activated carbon with optimum efficiency for cresol adsorption.

**Author Contributions:** A.S.: Conceptualization, Investigation, Methodology, Formal analysis, Writing—Original Draft, Writing—Review and Editing. U.B.S.: Writing—Review and Editing, Resources. D.S.: Supervision, Conceptualization, Resources, Writing—Review and Editing. All authors have read and agreed to the published version of the manuscript.

**Funding:** This research was funded by Department of Science and Technology, India. Grant no. IF160718.

**Institutional Review Board Statement:** Not applicable.

**Informed Consent Statement:** Not applicable.

**Data Availability Statement:** Not applicable.

**Acknowledgments:** The author Aola Supong is grateful to the Department of Science and Technology-INSPIRE Fellowship (IF160718). The authors are also grateful to Nikhil Guchhait, Department of Chemistry, Calcutta University, for extending the computational facility.

**Conflicts of Interest:** The authors declare no conflict of interest.

**Future Plans:** The results obtained in the theoretical studies suggested the favorability of carboxyl functionalized activated carbon. Based on these findings, future works may be planned to modify the surface of activated carbon by introducing carboxyl groups. Further, the synthesized carbon can be tested for its cresol removal efficiency experimentally. Similarly, carbonyl and hydroxyl functionalized activated carbons may be synthesized experimentally, and the studies on their removal efficiency/adsorption capacity will lead to the identification of a series of new and efficient activated carbons.

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
