# Peer review of "Density Functional Theory Calculations of the Effect of Oxygenated Functionals on Activated Carbon towards Cresol Adsorption"

_surfaces, doi:10.3390/surfaces5020020_

Round 1
Reviewer 1 Report
The paper presents a density functional theory study of the adsorption mechanism of cresol on different activated carbon models. The authors use an arm-chair model to prepresnt the activated carbon surface and compare the adsorption energy with the pristine and functionalized carbon models. They find that pristine activated carbon and introduction of carboxyls would be more favorable for cresol adsorption. Overall I find the paper interesting and comprehensive. Subject to considering the following queries, I recommend publication.
- In Materials and Methods section, the authors mention that B3LYP hybrid functional is applied for all calculations. To properly model van der Waals dispersion, usually a dispersion correction is added to DFT method. Please clarify whether the correction is added. It is crucial to obtain relatively accurate adsorption energy.
- “This model consisted of four fused benzene rings” - What is the rationale behind choosing this number of benzene rings? If 7 benzene rings are used, does the result change?
- How does the adsoprtion of cresol on pristine and functionalized activated carbon compare to the experimental reports?
- “Generally, adsorption energy of less than -30kJ/mol and more than 50 kJ/mol indicates physisorption and chemisorption respectively”. - Should the order of physisorption and chemisorption be swapped?
Reviewer 2 Report
Comments on surfaces-1489035
Title: Density functional theory calculations of the effect of oxygenated functionals on activated carbon towards cresol adsorption
Journal: Surfaces / MDPI
This manuscript was initiated which employed DFT calculations to study the interaction between activated carbon and p-cresol. The adsorption of p-cresol on pristine as well as functionalized activated carbon was studied. Different types of interactive mechanisms were studied, and the effect of three major functionals (hydroxyl, carboxyl, and carbonyl) on the p-cresol adsorption was accordingly investigated by the authors.
After a careful peer-reviewing process, I must inform you that, the subject of this paper is interesting and can be considered for publication in Surfaces after a MAJOR REVISION. I believe that the paper contains relevant information for the scientific community. I believe that the results are informative but must be well organized and improved in the next revision(s). Therefore, there are some questions about this submission and some revisions are necessary for this work. The major/minor issues are indicated as follows:
- The abstract is not well written. Some details of the experimental processes are missed in the abstract. Please revise this section.
- The state-of-the-art needs to be described more in “Introduction”. Please revise the last paragraph of this section.
- The new findings related to this work should be stated in the introduction clearly.
- Authors must show which questions/problems have been answered in this work. In this case, the correct and detailed information about the various methods/scopes of the density functional theory calculations must be provided in this manuscript.
- In this manuscript, results are well presented, but discussion on the obtained results must be completely provided in this manuscript. As can be seen, a “comprehensive” and “comparative” discussion specially on cresol adsorptions and relative adsorption energies are missed in this work. This should be provided as well.
- Please provide major findings in the ‘Conclusions” section with a bullet-point style.
- There is no description of the future plans for research in the first part of the “Conclusions” section. This should be completed in this section.
- Recently published references are beneficial for this work. Please check and use new references focused on your work.
- Also, please double-check and revise the reference list according to the Surfaces journal requirements.
- 3: Revise/enlarge the schemes and related text for better detection.
Round 2
Reviewer 2 Report
The revised manuscript looks fine and can be accepted now in Surfaces.
Congratulations!